# Glutamate Supplementation Ameliorated Growth Impairment and Intestinal Injury in High-Soya-Meal-Fed *Epinephelus coioides*

**DOI:** 10.3390/ani15162392

**Published:** 2025-08-14

**Authors:** Aozhuo Wang, Ruyi Xiao, Cong Huo, Kun Wang, Jidan Ye

**Affiliations:** Xiamen Key Laboratory for Feed Quality Testing and Safety Evaluation, Fisheries College of Jimei University, Xiamen 361021, China; 202212951071@jmu.edu.cn (A.W.); 201911908023@jmu.edu.cn (R.X.); 202211908038@jmu.edu.cn (C.H.); wangkun@jmu.edu.cn (K.W.)

**Keywords:** orange-spotted grouper, glutamate, growth performance, intestinal injury

## Abstract

Glutamate, the predominant non-essential amino acid in animals, is pivotal for protein biosynthesis and provides crucial oxidative energy for intestinal mucosal epithelial cells. This underpins its advantageous effects on the integrity of the intestinal barrier, antioxidant capacity, and protection against dietary stressor-induced enteritis (e.g., from anti-nutrition factors). However, the effectiveness of glutamate in reducing the negative effects of diets containing high-soya meal—specifically on growth and intestinal homeostasis in fish—is poorly understood. To address this gap, our study examined dietary glutamate supplementation as a means to prevent soya-meal-induced enteritis (SBMIE) in grouper *E*. *coioides*, elucidating its potential as a functional dietary strategy against SBMIE.

## 1. Introduction

Fish meal is broadly recognized as the primary high-grade protein ingredient in aquaculture feeds, particularly for marine and carnivorous freshwater fish. Its superiority stems from qualities like its high palatability, balanced profile of amino acids, and minimal anti-nutritional factors (ANFs) compared to other protein sources [1]. However, the expanding global aquaculture industry has intensified pressure on FM supplies [2], driving the search for alternative protein sources [3]. Soya meal (SBM) has emerged as the leading FM substitute due to its stable supply, relatively low cost, and favorable amino acid profile compared to other plant proteins [3,4,5].

A significant drawback of SBM, however, is its content of various ANFs, including antigenic proteins, soy lectins, and saponins. These compounds can induce soya-meal-type enteritis (SBMIE) in many farmed aquatic animals, especially carnivorous fish, when included at high dietary levels [6,7]. SBMIE impairs intestinal nutrient digestion and absorption, leading to reduced growth, poor feed utilization, and increased susceptibility to metabolic diseases [5]. Consequently, SBMIE represents not only a nutritional challenge but also a significant economic concern for aquaculture. Addressing this issue to mitigate its adverse effects on production is therefore critical. Over the past two decades, substantial research has focused on improving SBM utilization and controlling enteritis risk [3]. One promising strategy involves using functional feed additives to alleviate SBMIE and support intestinal homeostasis [8,9,10,11,12,13,14,15].

Glutamate (Glu), a prevalent non-essential amino acid in animals, contributes to protein biosynthesis [16,17,18], including in fish [19,20,21]. Dietary Glu supplementation enhances growth in fish [22,23,24,25,26], improves nutrient digestibility [27,28], and provides critical energy for intestinal mucosal epithelial cells via oxidation [29]. Beyond its nutritional role, Glu promotes intestinal cell proliferation [17], strengthens immune barrier function [30,31,32,33,34,35], and is considered a vital nutrient for these cells [36]. Furthermore, acting as a precursor for the biosynthesis of glutathione (GSH) and N-acetylglutamate within enterocytes [29], Glu exhibits pharmacological functions. It attenuates intestinal oxidative damage [27,30,37,38], improves intestinal integrity [39], and modulates the intestinal microbiota [9,40]. Collectively, these findings demonstrate Glu’s beneficial effects in maintaining intestinal barrier integrity, enhancing antioxidant capacity, and protecting against enteritis triggered by dietary stressors like ANFs [7]. Consequently, Glu supplementation is increasingly viewed as a nutritional solution path to mitigate the inflammatory reactions caused by high-plant-protein diets [39,40]. Despite this potential, research on Glu’s efficacy in improving growth and intestinal stress in high-SBM-fed fish diets remains limited [39,40,41,42,43].

Groupers, carnivorous marine fish extensively farmed in Southeast Asia [44], constitute a major mariculture industry in China, yielding a total aquaculture output of 241,480 tons in 2023 [45], among which the orange-spotted grouper (*Epinephelus coioides*) is the main aquaculture species after the hybrid grouper (*Epinephelus lanceolatus ♂ × E. fuscoguttatus ♀*). Research on the orange-spotted grouper confirms that high-SBM diets induce inflammatory responses and impair nutrient digestion and absorption, ultimately restricting growth [5,46]. Subsequent research by our team demonstrated that additives like sodium butyrate and lactoferrin can ameliorate SBMIE in this species [14,15]. However, the potential application of Glu as a functional additive in the intervention of SBMIE in groupers has not been explored yet. In this study, therefore, the purpose was to determine the efficacy of Glu supplementation in SBM diets in preventing SBMIE in this fish species. The findings will provide a basis for managing SBMIE in this commercially important cultured fish.

## 2. Materials and Methods

### 2.1. Test Feed and Rearing Management

A fish-meal-based diet (FM) containing 48% crude protein and 11% crude lipid was prepared. In the FM diet, 46% SBM was substituted for 30% FM to produce a high-SBM diet (SBM). Glu was incrementally supplemented at 1%, 2%, and 3% to the SBM diets and three test diets (marked as G-1, G-2, and G-3, respectively). The feed ingredients and nutrients are presented in Table 1. Following our established protocol [47], all five experimental diets were processed into 2.5 mm diameter pellet feed. Subsequently, the feed was dried at 55 °C for 24 h in a drying oven. Finally, the dry pellets were vacuum-sealed and cryopreserved in a −20 °C refrigerator until the growth trials.

The growth trial was carried out at the Mariculture Farm at Jimei University. Before starting the experiment, grouper juveniles (600 fish) were stocked in a concrete pond (L × W × H, 6 × 5 × 1.5 m) and hand-fed to satiation with the FM diet at each meal twice daily for a 3-week acclimatization period. The pond was cleaned every seven days. After cleaning, the sludge was discharged through the middle drainage hole at the bottom of the pond. The discharged sewage was then replenished with new seawater. In the initiation phase, 450 juvenile fish (mean initial weight: 15.1 g) underwent stochastic allocation to five experimental groups. Each group comprised three 500 L aquariums within a temperature-controlled recirculating aquaculture system (RAS), and 30 fish were kept in each aquarium. All aquariums were connected to a tap for the continuous supply of circulating water. The water flow rate of the tap was set to 8 L per minute. The standards (the feeding status, fish appearance, gill examination under the dissecting microscope, etc.) were used to determine if the fish were healthy. Juvenile groupers were fed their corresponding feeds by hand to satiation at each meal twice per day at 07:00 and 18:00 across an 8-week feeding duration. Postprandial management involved siphoning residual feed and fecal matter 30 min after each feeding event. Collected surplus feed was dried in a 65 °C oven for 24 h for mass quantification, enabling precise calculation of feed intake (FI). Water quality protocols included daily dissolved oxygen (DO) measurements at 15:00 h and biweekly nitrite-N analysis using a multiparameter photometer (HI83200, Hanna Instruments, Woonsocket, RI, USA). The rearing water temperature and DO and ammonia-N contents were recorded as 28 ± 0.5 °C, 6.0–7.5 mg/L, and 0.2 mg/L, respectively, across the entire trial.

### 2.2. Sample Collection

After the growth experiment was terminated, all the fish captured from each aquarium underwent chemical sedation in an immersion with 100 mg/L MS-222 (tricaine methanesulfonate; Sigma-Aldrich Shanghai Trading Co., Ltd., Shanghai, China). Anesthetized fish were subsequently batch-weighed with data recording conducted for performance metric calculations: weight gain (WG), specific growth rate (SGR), feed intake (FI), feed conversion ratio (FCR), and survival rate (SR). After measuring total fish count and weight in each aquarium, all the fish were placed back into their respective aquariums. Per-aquarium sampling involved five fish (anesthetic dosage of MS-222 described above). Individual biometric measurements preceded blood extraction from the tail vein using heparin-free syringes. Post-collection, the blood samples underwent 12 h coagulation at 4 °C prior to serum isolation (850× *g*, 4 °C, 10 min). Pooled serum aliquots (1.5 mL centrifuge tubes) were cryopreserved at −80 °C until the biochemical assay was performed. The same euthanized specimens were subsequently cut open for liver removal to determine hepatosomatic index (HSI) and condition factor (CF). The intestines (without digesta) and the dorsal muscles of fish from the same batch were aseptically excised and pooled into a 10 mL centrifuge tube by aquarium, then subsequently flash-frozen in liquid N_2_ and conserved in a −80 °C refrigerator for determining muscle composition, intestinal biochemical components, and mRNA levels of genes. Finally, three animals from each aquarium were randomly captured and merged by aquarium in polybags and conserved in a −20 °C refrigerator until the determination of nutrients was performed.

### 2.3. Nutrient and Amino Acid Analysis

The nutrients of the feed raw materials, the feeds, and the muscle and whole fish samples were analyzed using standard methods. Proximate composition analysis was conducted as follows: Dry matter was determined via a 105 °C drying oven to constant weight; total nitrogen quantified using the Kjeldahl method (Kjeltec System, Foss Tecator, Hoganas, Sweden); crude lipid extracted via a Soxtec Avanti 2050 (Foss Tecator) following the Soxhlet principle; and ash detected via 550 °C muffle furnace incineration for 8 h. Prior to analysis, samples underwent autoclave (121 °C, 20 min), homogenization, and 24 h dehydration at 65 °C for preparation.

As described in our previous study [48], the amino acids were assayed using an automatic amino acid analyzer (Hitachi L8900, Tokyo, Japan) after the hydrolysis of diets with 6-N HCl solution at 110 °C for 24 h.

### 2.4. Serum Component Determination

The serum total cholesterol (TC) and triglyceride (TG) contents were assayed via the GPO-PAP method; high-density lipoprotein cholesterol (HDL-C) and low-density lipoprotein cholesterol (LDL-C) were assayed via the peroxidase colorimetric method. The serum activities of alanine aminotransferase (ALT) and aspartate aminotransferase (AST) were measured via the Reitman–Frankel method; serum nitric oxide (NO) content was assayed via the nitrate reductase method, and urea nitrogen (BUN) content was measured via the diacetyl monoxime colorimetry method. All the serum components were determined by using their corresponding kits produced by Jiancheng Bioengineering Institute (Nanjing, China).

### 2.5. Assay for Digestive Enzyme Activity

The activities of intestinal amylase (the starch–iodine colorimetric assay), lipase (the methylresorufin substrate method), trypsin (ultraviolet colorimetry), and total protein (the Coomassie brilliant blue method) were assayed using kits (Nanjing Jiancheng Bioengineering Institute, Nanjing, China).

### 2.6. Assay for Antioxidant Capacity

The activities of intestinal glutathione (GSH) (the microplate assay method), superoxide dismutase (SOD) (the hydroxylamine method), total antioxidant capacity (T-AOC) (the colorimetric assay method), and catalase (CAT) (the ammonium molybdate method), as well as the malondialdehyde (MDA) levels (the thiobarbituric acid reaction method) were determined using their corresponding kits (Nanjing Jiancheng Bioengineering Institute).

### 2.7. Assay for Metabolic Enzyme Activity

The glutamine (Gln) synthetase (GS) (the colorimetric assay) and glutaminase (GLS) activity (the colorimetric assay) of intestinal samples were assayed with kits (Nanjing Jiancheng Bioengineering Institute).

### 2.8. Intestinal Histological Observation

One fish per aquarium was captured after the experiment was terminated. The intestine was collected, then cut into proximal, mid-, and distal parts (i.e., PI, MI, and DI). According to our previous methods described by Niu (2021) [48], segments were fixed with 4% paraformaldehyde. The fixed segments were processed using standard histological techniques with 70% (*v*/*v*) ethanol, followed by paraffin embedding treatment. Intestine sections (5 μm thickness) were mounted on glass slides and then underwent H and E staining. Histological evaluation utilized a Leica DM5500B light microscope (Wetzlar, Germany) at 100× magnification coupled with a DFC450 digital camera. Images were processed via Leica LAS AF 3.0 software (https://www.cellularimaging.nl/wp-content/uploads/LASAF30_ReleaseNotes_confocal.pdf (accessed on 6 June 2012)) for morphometric analysis of intestinal architecture, including height of the mucosal fold and muscle layer thickness (abbreviated as HMF and MLT).

### 2.9. RNA Extraction and mRNA Level Analysis

Intestinal total RNA was isolated via FastPure^®^ reagent (Novozan Biotech, Nanjing, China), with subsequent quantification of RNA concentration/purity using NanoDrop ND-2000 UV spectrophotometry and integrity verification through 1.2% agarose gel electrophoresis. The reverse transcription was performed with 1 μg of total RNA using TransScript^®^ reagent kit (Beijing Quanshijin Biotechnology Co., Ltd., Beijing, China). For the detection, we conducted quantitative real-time PCR (qRT-PCR) amplification of cDNA samples using ChamQ Universal SYBR qPCR Master Mix (Novozan Biotech, China) on an ABI 7500 system (Applied Biosystems, Foster, CA, USA). Reactions took place in the same detection system using SYBR Green chemistry (Toyobo, China) with gene-specific primers designed via Primer-BLAST (http://www.ncbi.nlm.nih.gov/tools/primer-blast/, accessed on 28 June 2025). Table 2 presents the sequences of primers.

Gene-specific primers (commercially provided with Suzhou Jinweizhi Biotech, Suzhou, China) underwent qPCR amplification under the following thermal profile: initial denaturation at 95 °C for 30 s; denaturation for 40 cycles at 95 °C for 5 s and annealing/extension at 60 °C for 30 s; and final dissociation. Amplification efficiency validation (~100%) [49] preceded gene expression quantification via the 2^−ΔΔCt^ method [50].

### 2.10. Statistical Analysis

Data are expressed as mean ± standard deviation (SD). Following verification of normality (Kolmogorov–Smirnov test) and homogeneity of variance (Levene’s test) in SPSS 22.0 (IBM, Armonk, NY, USA), all datasets underwent one-way ANOVA to detect inter-treatment differences. Percentage/ratio data were transformed prior to analysis. Regression modeling (excluding FM controls) assessed dose–response relationships between variables and dietary glutamate levels. Statistical significance was defined as *p* < 0.05.

## 3. Results

### 3.1. Growth Performance

Table 3 showed that WG and SGR were markedly lower in the SBM diet than in the FM diet (*p <* 0.05). Glu supplementation induced a dose-dependent increase in WG (r = 0.969) and SGR (r = 0.969), with maximum values observed in the G-3 diet that were restored to levels comparable to the FM diet (*p* > 0.05). Although no variations in FCR, FI, HSI, and CF were observed between the four treatments (*p >* 0.05), FCR and FI exhibited decreasing trends (r = 0.996 and r = 0.991 for FCR and FI, respectively), while CF showed an increasing tendency (r = 0.971) with increasing Glu levels.

### 3.2. Body and Muscle Nutrients

Table 4 did not demonstrate significant differences in the composition of the whole body/muscles across all treatments (*p* > 0.05).

### 3.3. Serum Biochemical Components

Figure 1 shows that serum HDL-C, TC, and AST levels in the SBM diet were markedly lower than that in the FM diet (*p <* 0.05). Glu supplementation promoted serum HDL-C content and AST activity in a dose-dependent pattern (r = 0.981 and r = 0.958 for HDL-C and AST, respectively) with increasing Glu levels, and diets G-2 and G-3 restored HDL-C and AST to the FM levels (*p >* 0.05), while serum TC remained unaffected by Glu supplementation (*p >* 0.05). In contrast, markedly higher LDL-C was observed in the SBM diet compared to the FM (*p <* 0.05). Glu supplementation reduced serum LDL-C with increasing Glu levels in a dose-dependent pattern (r = 0.946), and diets G-1 to G-3 normalized serum LDL-C levels to the FM levels (*p >* 0.05). Serum contents of TG, BUN, IgM, ALT, and DAO remained comparable across all four experimental groups (*p >* 0.05), although serum ALT (r = 0.958) and DAO (r = 0.921) levels followed the changing trend of HDL-C content, while serum TG (r = 0.928) and BUN (r = 0.947) contents followed the trend (r = 0.928 and r = 0.947) of LDL-C content.

### 3.4. Intestinal Antioxidant Capacity

Figure 2 displays that the SBM diet exhibited significantly reduced intestinal T-AOC, CAT, and SOD activities, but had an increased MDA level as compared to the FM diet (*p <* 0.05). Dietary Glu inclusion affected these antioxidant indices in a dose-dependent manner (r = 0.998 for T-AOC; r = 0.967 for CAT; r = 0.976 for SOD), achieving maximal values for T-AOC, CAT, and SOD and a minimum value for MDA in diet G-3. However, diets G-1 to G-3 normalized intestinal T-AOC and CAT levels to those of the SBM diet (*p >* 0.05), while diets G-2 and G-3 restored SOD and MDA to the levels of the FM diet (*p >* 0.05). Although intestinal GSH activity was unaffected by treatments (*p >* 0.05), there was an increasing trend of GSH with escalating Glu supplementation.

### 3.5. Intestinal Activities of Digestive Enzymes

Figure 3 displays that the intestinal activities of trypsin, lipase, and amylase were noticeably suppressed in the SBM diet in comparison to the FM diet (*p <* 0.05). Glu supplementation dose-dependently increased these digestive enzyme activities (r = 0.970, r = 0.965, and r = 0.987 for trypsin, lipase, and amylase, respectively), and Glu-added diets normalized intestinal trypsin and lipase to the FM levels (*p >* 0.05), while diet G-3 restored amylase to the FM level (*p >* 0.05).

### 3.6. Intestinal Activities and Gene Expression of Amino Acid Metabolic Enzymes

As shown in Figure 4, there is no significant difference in the intestinal GS activity between the SBM and FM diets (*p >* 0.05). However, dietary Glu supplementation induced a dose-dependent increase (r = 0.993) in intestinal GS activity, peaking in diet G-3 where values were noticeably higher than those of the FM diet (*p <* 0.05). In contrast, intestinal GLS activity was suppressed in the SBM diet in comparison with the FM diet (*p <* 0.05). Glu supplementation promoted intestinal GLS activity in a dose-responsive manner, with a maximum observed in diet G-3 that restored to the level of the FM diet (*p >* 0.05). The *Gls* mRNA level followed a similar pattern to GLS activity, with a maximum observed in diet G-3 that restored to the level of the FM diet (*p >* 0.05).

### 3.7. Intestinal Histomorphology

Table 5 demonstrates that the PI height of the mucosal fold was significantly reduced in the SBM versus the FM diet (*p <* 0.05). The PI HMF exhibited an increasing trend with Glu levels, but these values were similar to those of the SBM diet (*p >* 0.05), with maximum values reached at diet G-3 that did not restore to the FM level (*p >* 0.05). No significant differences were detected in muscularis thickness (PI) or in mucosal fold height/muscularis thickness (MI and DI) among treatments (*p >* 0.05).

As shown in Figure 5, those fed the FM diet exhibited well-developed mucosal folds with intact structure, densely arranged striated borders, thick muscularis, continuous serosa, and abundant goblet cells, whereas those fed the SBM diet displayed mucosal atrophy, severe fold disruption, curvature narrowing, and muscularis thinning. Comparatively, Glu supplementation ameliorated mucosal damage, with mid- and distal intestinal muscularis and lamina propria restored to the levels of the FM diet. Diet G-1 showed thickened but fragmented folds with a reduced quantity and thickened muscularis.

### 3.8. Intestinal Inflammatory Cytokine Gene Expression

Figure 6 shows that the mRNA levels of intestinal *IL-8*, *IL-12*, *IL-1β*, and *TNF-α* genes were significantly elevated, while the mRNA levels of intestinal *IL-10* and *TGF-β1* genes were noticeably suppressed in the SBM diet relative to the FM diet (*p <* 0.05). Glu supplementation dose-dependently downregulated the mRNA levels of intestinal pro-inflammatory factor genes (*IL-8*, *IL-12*, *IL-1β,* and *TNF-α* for r = 0.914, r = 0.894, r = 0.890, and r = 0.945, respectively), reaching minimal levels in the G-3 group that were statistically equivalent to the FM values (*p >* 0.05). In contrast, dietary Glu supplementation progressively upregulated the mRNA levels of anti-inflammatory factor genes (*IL-10* and *TGF-β1*) (r = 0.930 and r = 0.915, respectively), achieving maximal values in diet G-3 comparable to FM levels (*p >* 0.05).

## 4. Discussion

Growth in carnivorous and marine fish is often constrained by ANFs and amino acid imbalances when high levels of SBM replace FM in feeds [7,43]. As an effective functional nutrient, Glu supplementation has been demonstrated to enhance growth, modulate oxidative stress, and support immune function in fish under both stressed and non-stressed conditions [13,35,36,51]. In the current study, the linear improvement in WG and SGR with up to 3% dietary Glu indicates that this dosage achieved maximal growth restoration, comparable to FM-based diets. Non-significant trends of decreasing FCR and FI with increasing Glu suggest potential efficiency gains at the 3% inclusion level. Although HSI and CF were unaffected by Glu supplementation, their patterns mirrored growth rates. Similar dose-dependent effects have been reported in rainbow trout [17] and tilapia [26]. Furthermore, numerous studies consistently report positive effects of Glu supplementation on the growth and feed utilization of high-SBM-fed fish [7,17,22,24,35,39,41,42,52,53,54]. However, the optimal Glu level varies considerably, ranging from 0.5% [37] to 6% [55], likely reflecting fish differences, growth stages, and feed composition and formulations using alternative protein sources in low-FM diets [26]. On the other hand, some reports have found that excessive Glu reduced fish growth, possibly due to metabolic disorders caused by amino acid imbalance [56,57].

Notably, Glu plays a fundamental role in regulating growth and maintaining intestinal integrity [33]. The observed growth enhancement aligns with improved nutrient utilization efficiency in fish receiving Glu-supplemented diets [28,53], corroborated by the increased activities of intestinal digestive enzymes observed in this study and previous research [21,55,58,59,60]. This improved digestive capacity corresponds with the Glu-enhanced intestinal integrity (e.g., increased villus height, goblet cell numbers) observed here and elsewhere [17,26,39,42], attributed to Glu maintaining the energy supplement for intestinal mucosal cells [28,33,52,60]. In groupers experiencing SBM-induced intestinal histomorphological damage in this study, Glu supplementation dose-dependently restored mucosal architecture. Specifically, 3% Glu yielded optimal fold integrity characterized by elongated and densely arranged folds, concomitant with improved intestinal digestion, facilitating better nutrient absorption [53].

Additionally, Glu supplementation elevated serum HDL-C and reduced LDL-C compared to SBM diets, restoring them to levels equivalent to the FM group. This suggests a partial mitigation of SBM-induced dyslipidemia, consistent with Glu’s role in modulating lipid metabolism in other species like pigs [61]. Concurrently, elevated AST activities in fish fed 2–3% Glu versus SBM diets may indicate enhanced transamination, potentially promoting amino acid deposition in muscle [20]. Enhanced amino acid metabolism is often associated with increased plasma Glu levels [62]. Serum ALT, BUN, and DAO levels (non-significant) followed trends similar to HDL-C, implying impacts on hepatic and intestinal functions [14,15]. These improvements in nutritional status may partially explain how Glu alleviates SBM-induced growth retardation.

Notably, Glu supplementation dose-dependently enhanced intestinal antioxidant capacity, evidenced by increased intestinal SOD and CAT activities, and T-AOC level, alongside reducing MDA levels. This indicates the mitigation of intestinal oxidative damage, although CAT and T-AOC in Glu-fed fish did not fully recover to FM levels. These findings align with previous reports that Glu enhances reactive oxygen species scavenging and reduces oxidative injury in fish [13,37,54]. Glu has also been shown to protect against oxidative stress induced by soy saponins or LPS/H_2_O_2_ in fish intestinal epithelial cells [7,35,36]. While intestinal SOD recovered to FM levels, the incomplete restoration of T-AOC and CAT suggests that higher Glu doses or additional pathways, such as enhanced α-ketoglutarate production, may be necessary. α-ketoglutarate enters the tricarboxylic acid cycle, generating intermediates like fumarate and oxaloacetate that chelate iron ions to inhibit lipid peroxidation [63]. Increased intestinal GSH levels, previously reported in non-stressed fish fed Glu [26,27], showed a dose-dependent (non-significant) increase in the SBM-stressed groupers in this study. Rising GSH supports the role of Glu as a precursor for GSH biosynthesis [58,64,65], contributing to reduced lipid peroxidation. Furthermore, the Glu/Gln cycle, mediated by Gln synthetase (GS) and Glu synthase (GLS), is crucial for intestinal nitrogen metabolism [32,66]. As a Gln precursor, Glu supports immune function and protein synthesis [29,67]. Our data revealed dose-dependent upregulation of GS/GLS activities and *Gls* expression with increasing dietary Glu, with levels restored to those of FM at 3% Glu. This echoes the high GLS activity and *GS/Gls* expression reported in LPS/glycinin-stressed fish [33,40], suggesting that high-SBM diets increase Gln demand, necessitating higher Glu supplementation than FM-based diets.

Simultaneously, Glu supplementation dose-dependently downregulated the mRNA levels of pro-inflammatory cytokine genes in the intestine (*IL-8*, *IL-12*, *IL-1β*, *TNF-α*) while upregulating the mRNA levels of anti-inflammatory cytokine genes in the intestine (*IL-10*, *TGF-β1*) in groupers. Similarly, Gln supplementation alleviated intestinal inflammation in SBM-fed turbot [68]. The balance between controlled inflammation and mucosal epithelial recovery is critical for fish health [23,36], indicating Glu’s contribution to ameliorating SBM-induced intestinal injury. Importantly, 3% Glu supplementation normalized the intestinal inflammatory response to FM levels, suggesting that SBM-fed fish require higher Glu intake than non-stressed fish to sustain intestinal homeostasis [69]. The results from this experiment confirm that dietary Glu supplementation promotes growth performance in SBM-fed groupers in a dose-dependent pattern, achieving restoration comparable to FM levels at 3% Glu. This growth improvement corresponds with the effective alleviation of SBM-induced liver and intestinal damage. Specifically, Glu supplementation enhanced intestinal digestion capacity and integrity and antioxidant capacity while elevating resistance to enteritis.

## 5. Conclusions

The results of the current study show that adding Glu to feed promotes growth performance in SBM-fed groupers in a dose-dependent pattern, achieving restoration comparable to the FM levels at 3% Glu. This growth improvement corresponds with the effective alleviation of SBM-induced liver and intestinal damage. Specifically, Glu supplementation enhanced intestinal digestion capacity and integrity and antioxidant capacity while elevating resistance to enteritis.

## Figures and Tables

**Figure 1 animals-15-02392-f001:**
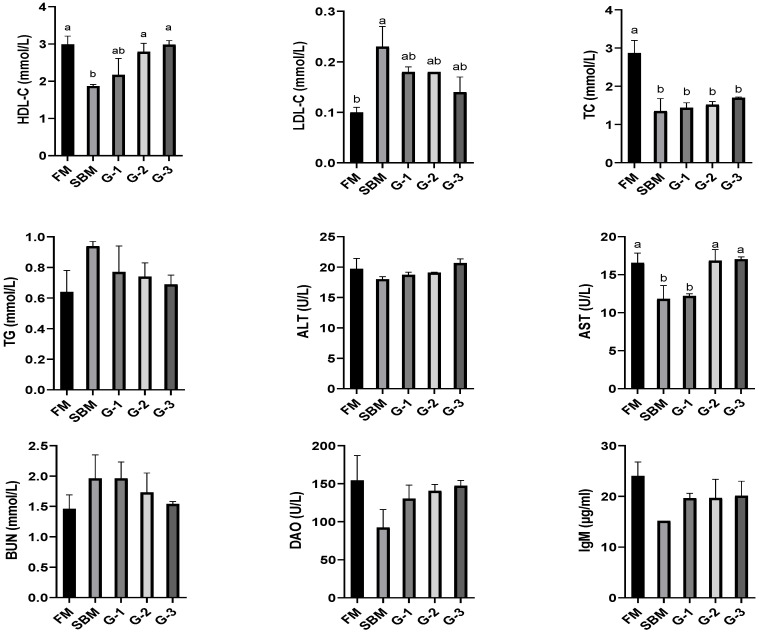
Effects of glutamate addition on serum biochemical components of high-soya-meal-fed orange-spotted groupers. Bars with differing lowercase letters indicate significant differences (*p* < 0.05). FM, diet containing 52% fish meal; SBM, diet containing 46% soya meal; diets G-1 to G-3, SBM diets with 1, 2, or 3% glutamate addition. TC, total cholesterol; TG, triglyceride; HDL-C, high-density lipoprotein cholesterol; LDL-C, low-density lipoprotein cholesterol; ALT, alanine aminotransferase; AST, aspartate aminotransferase; DAO, diamine oxidase; BUN, blood urea nitrogen; IgM, immunoglobulinM.

**Figure 2 animals-15-02392-f002:**
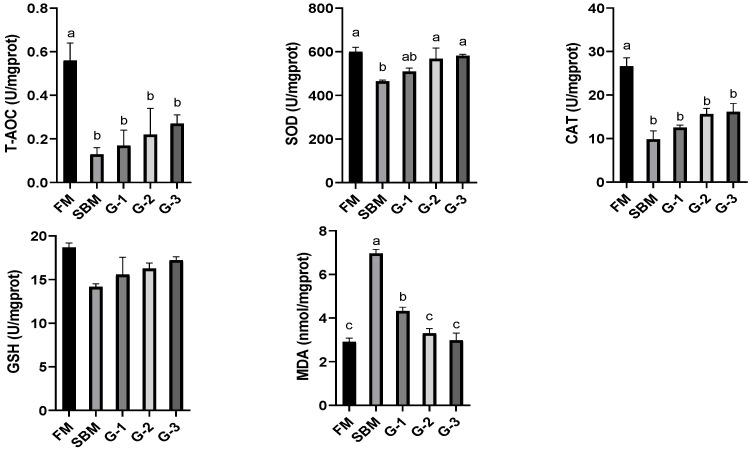
Effects of glutamate addition on intestinal antioxidant capacity of high-soya-meal-fed orange-spotted groupers. Bars with differing lowercase letters indicate significant differences (*p* < 0.05). FM, diet containing 52% fish meal; SBM, diet containing 46% soya meal; diets G-1 to G-3, SBM diets with 1, 2, or 3% glutamate addition. T-AOC, total antioxidant capacity; SOD, superoxide dismutase; CAT, catalase; GSH, glutathione; MDA, malondialdehyde.

**Figure 3 animals-15-02392-f003:**
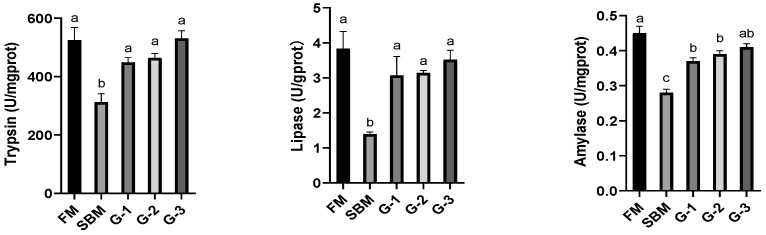
Effects of glutamate addition on the activities of intestinal digestive enzymes of high-soya-meal-fed orange-spotted groupers. Bars with differing lowercase letters indicate significant differences (*p* < 0.05). FM, diet containing 52% fish meal; SBM, diet containing 46% soya meal; diets G-1 to G-3, SBM diets with 1, 2, or 3% glutamate addition.

**Figure 4 animals-15-02392-f004:**
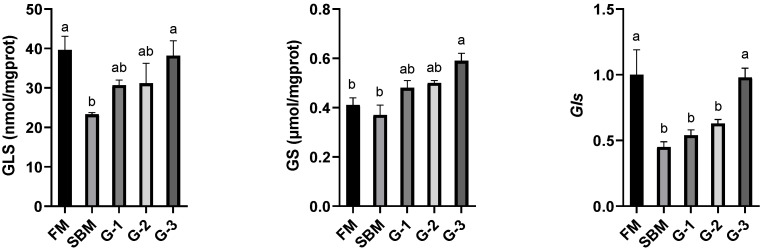
Effects of glutamate addition on the activities and gene mRNA levels of amino acid metabolic enzymes of high-soya-meal-fed orange-spotted groupers. Bars with differing lowercase letters indicate significant differences. FM, diet containing 52% fish meal; SBM, diet containing 46% soya meal; diets G-1 to G-3, SBM diets with 1, 2, or 3% glutamate addition. GS, glutamine synthetase; GLS, glutaminase. *Gls*, glutaminase gene.

**Figure 5 animals-15-02392-f005:**
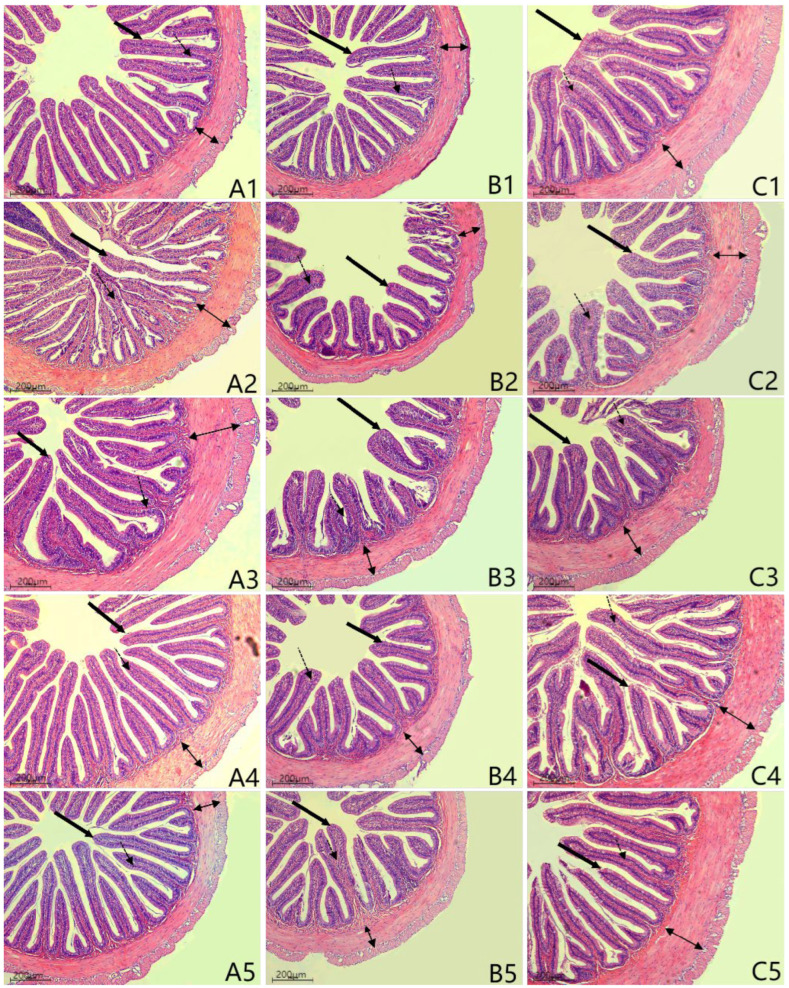
Histology in proximal, mid-, and distal intestines of high-soya-meal-fed orange-spotted groupers 100×. FM, diet containing 52% fish meal; SBM, diet containing 46% soya meal; diets G-1 to G-3, SBM diets with 1, 2, or 3% glutamate addition. (**A**–**C**) are the proximal, mid-, and distal intestines, respectively; 1, 2, 3, 4, and 5 represent groups FM, SBM, G-1, G-2, and G-3, respectively; the black bold single arrow means goblet cells; the black two-way arrow means the thickness of the muscle layer; the black thin single arrow means the lamina propria.

**Figure 6 animals-15-02392-f006:**
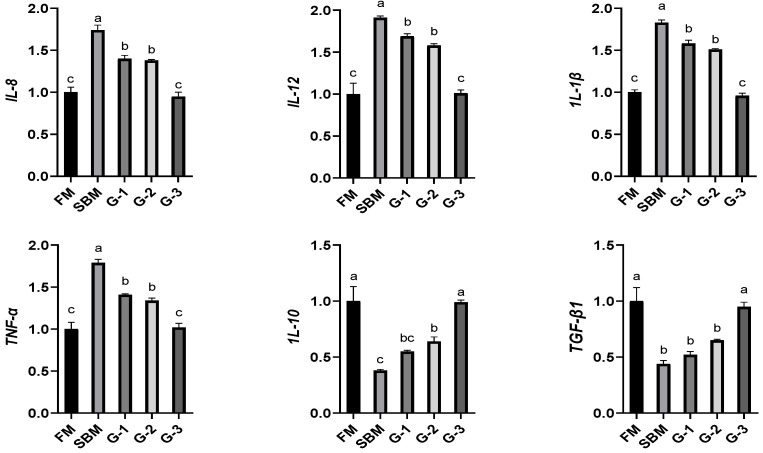
Effects of glutamate addition on mRNA levels of intestinal inflammatory factor genes of high-soya-meal-fed orange-spotted groupers. Bars with differing lowercase letters indicate significant differences (*p* < 0.05). FM, diet containing 52% fish meal; SBM, diet containing 46% soya meal; diets G-1 to G-3, SBM diets with 1, 2, or 3% glutamate addition. IL, interleukin; TNF-α, tumor necrosis factor-α; TGF-β1, transforming growth factor-β1.

**Table 1 animals-15-02392-t001:** Ingredients and nutrients of experimental diets (dry matter basis).

Ingredients (g/kg)	Diets				
	FM	SBM	G1	G2	G3
Fish meal	520.0	220.0	220.0	220.0	220.0
Casein	131.0	120.0	120.0	120.0	120.0
Gelatin	33.0	30.0	30.0	30.0	30.0
Soya meal	0	460.0	460.0	460.0	460.0
Soya oil	42.6	42.6	42.6	42.6	42.6
Fish oil	8.2	30.8	30.8	30.8	30.8
Soy lecithin	20.0	20.0	20.0	20.0	20.0
Added glutamate	0	0	10.0	20.0	30.0
Corn starch	206.8	38.3	28.3	18.3	8.3
Sodium alginate	10.0	10.0	10.0	10.0	10.0
Ca(H_2_PO_4_)_2_	15.0	15.0	15.0	15.0	15.0
Choline chloride	4.0	4.0	4.0	4.0	4.0
Stay-C 35%	0.3	0.3	0.3	0.3	0.3
Premix	9.0	9.0	9.0	9.0	9.0
Nutrient level					
Dry matter (g/kg)	905.8	907.8	903.1	905.5	905.8
Crude protein (g/kg)	484.0	482.9	493.2	491.0	501.9
Crude lipid (g/kg)	114.1	120.6	116.7	117.9	120.5
Ash (g/kg)	97.5	79.3	80.6	80.9	80.9
Total glutamate (g/kg)	61.2	73.1	82.2	93.1	104.5
Gross energy (MJ/kg)	21.17	21.44	21.32	21.33	21.27

FM, diet containing 52% fish meal; SBM, diet containing 46% soya meal; diets G-1 to G-3, SBM diets with 1, 2, or 3% glutamate addition. Fish meal, soya meal, and soya oil were obtained from Jiakang Feed Co. Ltd., Xiamen, China; Stay-C 35% was provided by Shanghai Yuanye Bio-Technology Co., Ltd., Shanghai, China; glutamate was provided by Shanghai Acmec Biochemical Technology Co., Ltd., Shanghai, China; gelatin and casein were provided by Gansu Hualing Dairy Co., Ltd., Hezuo, China; choline chloride was provided by Shandong Yinglang Chemical Co., Ltd., Jinan, China; sodium alginate was provided by Qingdao Yifang Chemical Co., Ltd., Qingdao, China; soya lecithin was purchased from Fujian Juhai Biotechnology Co., Ltd., Nanan, China; Ca (H_2_PO_4_)_2_ was provided by Jinan Lusheng Chemical Co., Ltd., Jinan, China; corn starch was provided by Hebei Derui Starch Co., Ltd., Xinji, China. The gross energy values of feeds were calculated based on the complete combustion calorific values of fat (39.5 MJ/kg), protein (23.6 MJ/kg), and carbohydrates (17.2 MJ/kg). The vitamin and trace mineral composition of premix was provided as specified in Qin (2022) [47].

**Table 2 animals-15-02392-t002:** The sequences of primers that were used for real-time PCR.

Genes	Forward Primer (5′-3′)	Reverse Primer (5′-3′)	E (%)	Accession No.
*IL-8*	AAGTTTGCCTTGACCCCGAA	TGAAGCAGATCTCTCCCGGT	94	FJ9130641
*IL-1β*	GCAACTCCACCGACTGATGA	ACCAGGCTGTTATTGACCCG	116	EF582837.1
*IL-10*	GTCCACCAGCATGACTCCTC	AGGGAAACCCTCCACGAATC	99	KJ741852.1
*TGF-β1*	GCTTACGTGGGTGCAAACAG	ACCATCTCTAGGTCCAGCGT	102	GQ503351.1
*IL-12*	CCAGATTGCACAGCTCAGGA	CCGGACACAGATGGCCTTAG	115	KC662465.1
*TNF-a*	GGATCTGGCGCTACTCAGAC	CGCCCAGATAAATGGCGTTG	117	FJ009049.1
*Gls*	CCCGAACATACGACCAGAGG	CATGATGCGCACTTAGCCAC	111	CL1126.Cotig86_All
*β-actin*	GATCTGGCATCACACCTTCT	CATCTTCTCCCTGTTGGCTT	104	AY510710.2

IL, interleukin; TGF, transforming growth factor; TNF, tumor necrosis factor; *Gls*, glutaminase gene.

**Table 3 animals-15-02392-t003:** Effects of glutamate addition on the growth performance of high-soya-meal-fed orange-spotted groupers.

Parameters	Diets				
	FM	SBM	G-1	G-2	G-3
IABW (g)	15.10 ± 0.02	15.11 ± 0.01	15.11 ± 0.04	15.10 ± 0.01	15.09 ± 0.03
FABW (g)	81.93 ± 1.32 ^a^	65.87 ± 0.63 ^b^	68.00 ± 4.59 ^b^	75.25 ± 4.06 ^ab^	77.16 ± 1.58 ^ab^
WG (%)	442.72 ± 8.81 ^a^	335.95 ± 3.91 ^b^	349.81 ± 29.38 ^b^	398.42 ± 27.02 ^ab^	411.43 ± 11.36 ^ab^
SGR (%/d)	3.02 ± 0.03 ^a^	2.63 ± 0.02 ^b^	2.68 ± 0.12 ^b^	2.86 ± 0.10 ^ab^	2.91 ± 0.04 ^ab^
FCR	0.95 ± 0.01	1.19 ± 0.09	1.13 ± 0.10	1.05 ± 0.03	1.00 ± 0.05
FI (%/d)	2.34 ± 0.02	2.66 ± 0.19	2.55 ± 0.17	2.50 ± 0.10	2.41 ± 0.12
HSI (%)	1.95 ± 0.16	1.33 ± 0.19	1.52 ± 0.33	1.27 ± 0.07	1.63 ± 0.03
CF (%)	2.89 ± 0.05	2.67 ± 0.03	2.77 ± 0.06	2.83 ± 0.07	2.86 ± 0.08
SR (%)	98.89 ± 1.11	96.67 ± 1.93	98.89 ± 1.11	98.89 ± 1.11	98.89 ± 1.11

Significance difference is indicated when data have different superscripts in the same row (*p* < 0.05). FM, diet containing 52% fish meal; SBM, diet containing 46% soya meal; diets G-1 to G-3, SBM diets with 1, 2, or 3% glutamate addition. IABW, initial average body weight (g/fish); FABW, final average body weight (g/fish); WG, weight gain (%) = 100 × (FABW − IABW)/IABW; SGR, specific growth rate (%/d) = 100 × (ln FABW − ln IABW)/days; FCR, feed conversion rate = FI/(FABW − IABW); FI, feeding intake (%/d) = 100 × FABW/(FABW + IABW)/2/days; HSI, hepatosomatic somatic index (%) = 100 × liver weight (g)/whole-body weight (g/fish); CF, condition factor (%) = 100 × whole-body weight (g/fish)/(whole-body length (cm)^3^); SR, survival rate (%) = 100 × final count of fish/initial count of fish.

**Table 4 animals-15-02392-t004:** Effects of glutamate addition on the composition of the whole body and muscles of high-soya-meal-fed orange-spotted groupers (%, wet-weight basis).

Parameters (%)		Diets				
		FM	SBM	G-1	G-2	G-3
Whole body	Moisture	69.46 ± 0.23	69.12 ± 0.01	69.89 ± 0.42	69.94 ± 0.32	69.23 ± 0.36
	Crude protein	16.55 ± 0.19	17.17 ± 0.14	17.10 ± 0.37	16.28 ± 0.35	17.16 ± 0.18
	Crude lipid	7.45 ± 0.09	7.23 ± 0.15	7.33 ± 0.29	7.36 ± 0.53	7.67 ± 0.27
	Ash	4.54 ± 0.12	4.41 ± 0.03	4.32 ± 0.05	4.35 ± 0.09	4.40 ± 0.12
Muscle	Moisture	75.65 ± 0.72	74.52 ± 0.26	74.90 ± 0.24	75.00 ± 0.33	75.54 ± 0.55
	Crude protein	19.38 ± 0.81	20.57 ± 0.04	20.70 ± 0.12	20.27 ± 0.06	19.67 ± 0.39
	Crude lipid	2.97 ± 0.46	2.65 ± 0.36	2.24 ± 0.23	2.33 ± 0.48	2.30 ± 0.20
	Ash	1.24 ± 0.06	1.36 ± 0.01	1.32 ± 0.01	1.30 ± 0.00	1.22 ± 0.06

No significant difference is indicated when data have no letter superscripts in the same row (*p* > 0.05). FM, diet containing 52% fish meal; SBM, diet containing 46% soya meal; diets G-1 to G-3, SBM diets with 1, 2, or 3% glutamate addition.

**Table 5 animals-15-02392-t005:** Effects of glutamate addition on the intestinal morphometric measurements of high-soya-meal-fed orange-spotted groupers.

Parameters		Diets				
		FM	SBM	G-1	G-2	G-3
PI	HMF (μm)	724.63 ± 21.20 ^a^	434.34 ± 18.26 ^b^	434.80 ± 18.03 ^b^	537.18 ± 85.36 ^b^	585.92 ± 16.44 ^b^
	MT (μm)	243.39 ± 23.53	159.31 ± 8.54	166.12 ± 4.94	176.07 ± 27.11	218.76 ± 29.34
MI	HMF (μm)	587.18 ± 35.55	388.78 ± 79.70	394.87 ± 39.67	445.47 ± 16.89	513.48 ± 47.04
	MT (μm)	209.82 ± 19.10	128.23 ± 46.37	129.03 ± 13.36	170.58 ± 4.61	192.14 ± 21.55
DI	HMF (μm)	508.90 ± 30.65	447.21 ± 11.21	481.29 ± 63.97	484.80 ± 86.20	500.20 ± 95.76
	MT (μm)	217.13 ± 37.45	154.86 ± 19.38	211.38 ± 40.56	211.65 ± 30.54	212.95 ± 37.56

Significance difference is indicated when data have different superscripts in the same row (*p* < 0.05). FM, diet containing 52% fish meal; SBM, diet containing 46% soya meal; diets G-1 to G-3, SBM diets with 1, 2, or 3% glutamate addition. PI, proximal intestine; MI, mid-intestine; DI, distal intestine; HMF, height of mucosal fold; MT, muscle thickness.

## Data Availability

The data that support the conclusions of this article will be available from the corresponding author upon reasonable request.

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
