# Peer review of "Glutamate Supplementation Ameliorated Growth Impairment and Intestinal Injury in High-Soya-Meal-Fed Epinephelus coioides"

_animals, 2025, doi:10.3390/ani15162392_

Round 1
Reviewer 1 Report
Comments and Suggestions for Authors
The authors present in this paper a study aimed at evaluating the efficacy of dietary glutamate (Glu) supplementation in counteracting growth retardation and intestinal stress injury in juvenile orange-spotted grouper.
The introduction is well-documented, well-argued, and provides a clear conceptual framework for the proposed research.
However, I believe the introduction should also include some information about the potential limitations of glutamate use, such as costs, possible side effects at high doses, etc.
Based on the literature cited in the introduction, the authors identify a knowledge gap regarding the effect of glutamate on orange-spotted grouper (Epinephelus coioides), a species of significant economic importance in Asia. This fully justifies the research presented in the current manuscript.
The "Materials and Methods" chapter is well written, offering a clear picture of how the study was prepared and conducted.
On line 122, the phrase “…coefficient ratio” should be replaced with “…conversion ratio” or “…conversion rate”. The same applies to line 227.
The "Results" chapter contains valid data and valuable information, well described. However, I have a few suggestions:
I suggest moving the formulas listed below Table 3 into the "Materials and Methods" chapter (lines 122–123 and 129–130). Only the figure legend should remain below Table 3.
The figures on lines 253 and 272 contain bar graphs without letters above the bars. The legend should clearly state what these represent.
Lines 255 and 256 repeat the sentence “Bars bearing different letters are significantly different (P < 0.05).” One of them should be removed.
The "Discussion" chapter is well written.
The "Conclusions" are pertinent and supported by the experimental data.
After minor revision, the paper may be suitable for publication.
Author Response
Reviewer 1
The authors present in this paper a study aimed at evaluating the efficacy of dietary glutamate (Glu) supplementation in counteracting growth retardation and intestinal stress injury in juvenile orange-spotted grouper.
The introduction is well-documented, well-argued, and provides a clear conceptual framework for the proposed research.
Response:We sincerely thank the reviewer for their positive evaluation of our introduction and the overall conceptual framework. We are encouraged by your recognition of our efforts to contextualize this study within existing literature.
However, I believe the introduction should also include some information about the potential limitations of glutamate use, such as costs, possible side effects at high doses, etc.
Response:Thank you for your valuable suggestions. But we think it would be more appropriate to place this topic in the discussion section.
Based on the literature cited in the introduction, the authors identify a knowledge gap regarding the effect of glutamate on orange-spotted grouper (Epinephelus coioides), a species of significant economic importance in Asia. This fully justifies the research presented in the current manuscript.
The "Materials and Methods" chapter is well written, offering a clear picture of how the study was prepared and conducted.
Response:Many thanks to your encouraging comments.
On line 122, the phrase “…coefficient ratio” should be replaced with “…conversion ratio” or “…conversion rate”. The same applies to line 227.
Response:Thanks to your reminding of us, this error was corrected.
The "Results" chapter contains valid data and valuable information, well described. However, I have a few suggestions:
I suggest moving the formulas listed below Table 3 into the "Materials and Methods" chapter (lines 122–123 and 129–130). Only the figure legend should remain below Table 3.
Response:Thank you very much for your kind suggestion. Despite of this, we determine to keep the arrangement without adjustment. This practice is beneficial to directly introduce the readers to understand the experimental results from the table.
The figures on lines 253 and 272 contain bar graphs without letters above the bars. The legend should clearly state what these represent.
Response:We have supplemented the explanation regarding the superscript letters above the bars.
Lines 255 and 256 repeat the sentence “Bars bearing different letters are significantly different (P < 0.05).” One of them should be removed.
Response: The repeat sentence has been removed.
The "Discussion" chapter is well written.
The "Conclusions" are pertinent and supported by the experimental data.
Response:Thanks to your encouraging comments.
After minor revision, the paper may be suitable for publication.
Reviewer 2 Report
Comments and Suggestions for Authors
The subject of the manuscript is relevant, the research contributes significant elements to the state of the art in the research species (Epinephelus coioides), however the discussion lacks clarity, and a lack of information is detected in some segments of the document, to have a clearer vision, suggested changes in the manuscript are required:
Comments for authors:
A graphic abstract with an experimental design is required.
Line 89; the diets varied in their protein and lipid composition, what are the reasons for these variations?
Line 94; feed characteristics (type, diameter or diameters if different sizes were used), must be included.
Line 97; what was the purpose of using soybean oil, soybean lecithin, and fish oil in the diets?
Why were gelatin and sodium alginate used in the feeds? apparently both are binders?
The energy and hydrostability values ​​of the feeds are missing.
The FAN analysis of the feeds is missing.
The composition of the premix must be attached.
Because the standard deviation of the proximate chemical composition is not presented.
Ingredients and nutrients must be specified on Table 1 (dry or wet basis).
The type of premix must be specified, in their individual contributions.
The brand name of the ingredients is required (vitamin C, glutamate, choline chloride, sodium alginate, gelatin, casein, soybean lecithin, calcium, corn starch).
Lines 104-106; attach the acclimatization conditions of the fish, type of Feeding, ration type, number of fish in the initial population.
It is required to attach the bioassay feeding rate, timing of biometrics (if any), water exchange levels, water flow rate in the ponds, how fish health was assessed, type of photoperiod, how mortality was estimated, type of heaters used, type of physical or chemical treatments applied to the water (if applicable), what animal welfare protocol was applied to the management of the system with the fish.
Lines 119-212; are confusing: the number of fish anesthetized. Is there duplicate information?
Line 124; were heparin syringes used? If so, this should be described.
Line 126; set the rpm to g.
Lines 133-135; are confusing because they appear to indicate that all fish in the experiment were anesthetized. Verify the information.
Lines 151-157-162-164; mention the kit´s description.
Line 174; delete the period.
Lines 207-209; this description regarding the way the results are represented raises questions. Why do the figures and the results description compare the 5 treatments? Clarify. If the data in Figures 1 through 4 are in the same figure, these data should be comparative with respect to the control and with each other.
Line 216; adjust the comment to the statistical comparison: "There are no differences between these 3 treatments."
In Table 3, how did mortality influence the calculations of feed consumed and FCR? If there was an adjustment, it should be mentioned.
Line 242; the letter r should be added.
In Figures 1, 2, 3, and 4, the highest value should be marked with the letter a, the lowest value with the letter b, and so on.
Line 255; is missing a parenthesis.
Line 256; repeats the previous sentence.
Figure captions 3 and 4, standardize them with the format of Figures 1 and 2.
On Line 301; verify that the term "mucosal fold height" corresponds to the initials in Table 5 and is consistent with the description in the footnote.
On Lines 301-302; verify that only two treatments, SBM and FM, are compared; however, the other treatments are not mentioned (with the same response as SBM).
In Table 5, standardize the format in the footnotes.
In Figure 6, add the measurement bar in the corresponding unit.
On Lines 328 and 332; standardize the P format of the document.
On Line 330; add the letter r to each piece of data.
On Line 336; standardize the footnotes with those of the other figures.
I suggest separating the discussion into sections based on the order of the results with their respective headings.
On Line 456; verify that the journal appears as Fisheries and Aquatic Science
Comments on the Quality of English LanguageBased on the suggested changes, the presentation of the text must be adapted and revised in English.
Author Response
Reviewer 2
(1) The subject of the manuscript is relevant, the research contributes significant elements to the state of the art in the research species (Epinephelus coioides), however the discussion lacks clarity, and a lack of information is detected in some segments of the document, to have a clearer vision, suggested changes in the manuscript are required:
Response: Thank you for your encouraging and valuable comments.
Comments for authors:
(2) A graphic abstract with an experimental design is required.
Response: GA was inserted into the text.
(3) Line 89; the diets varied in their protein and lipid composition, what are the reasons for these variations?
Response: This is normal because there are certain errors in the weighing of each ingredient based upon the feed formulations when preparing feed, and there are also certain errors in the chemical analysis of every experimental diet. The given values of protein and lipid contents are only the result of feed formula calculations, not analyzed values.
(4) Line 94; feed characteristics (type, diameter or diameters if different sizes were used), must be included.
Response: This was done.
(5) Line 97; what was the purpose of using soybean oil, soybean lecithin, and fish oil in the diets?
Response: The use of soybean oil is for the purpose of reducing the use of fish oil while ensuring sufficient supply of essential fatty acids. The purpose of using soybean lecithin is to provide sufficient phospholipids for experimental fish.
(6) Why were gelatin and sodium alginate used in the feeds? apparently both are binders?
Response: In this study, gelatin is used as a protein source despite its strong adhesive property. In the study of fish nutrition, casein and gelatin are often used in combination as high-quality animal protein sources.
(7) The energy and hydrostability values of the feeds are missing.
Response: We provided gross energy of the diets. Unfortunately, the water stability of the feed has not been determined and cannot be provided. However, the presence of adhesive components (gelatin and sodium alginate) in the feed ensures that the prepared feed can maintain for at least 2 hours without diffusion based on our past feed preparation practices.
(8) The FAN analysis of the feeds is missing.
Response: We did not determine free amino nitrogen (FAN) of experimental diets. In the study, the protein ingredients used in the formulated feed contain almost no free amino acids and other non-protein nitrogen, except for specially added Glu. For this reason, the significance of determining FAN is relatively small.
(9) The composition of the premix must be attached.
Response: The composition of the premix was provided by our team’s Qin (2022) to reduce redundancy.
(10) Because the standard deviation of the proximate chemical composition is not presented.
Response: Normally, the standard deviation of proximate composition of feed is not listed. The composition of feed determines the success or failure of growth experiments. The accuracy of the composition of experimental feed is important.
(11) Ingredients and nutrients must be specified on Table 1 (dry or wet basis).
Response: This was done as suggested by the reviewer.
(12) The type of premix must be specified, in their individual contributions.
Response: The premix was specified in the footnote.
(13) The brand name of the ingredients is required (vitamin C, glutamate, choline chloride, sodium alginate, gelatin, casein, soybean lecithin, calcium, corn starch).
Response: We added the brand names of these ingredients.
(14) Lines 104-106; attach the acclimatization conditions of the fish, type of Feeding, ration type, number of fish in the initial population.
Response: We supplemented the acclimatization conditions.
(15) It is required to attach the bioassay feeding rate, timing of biometrics (if any), water exchange levels, water flow rate in the ponds, how fish health was assessed, type of photoperiod, how mortality was estimated, type of heaters used, type of physical or chemical treatments applied to the water (if applicable), what animal welfare protocol was applied to the management of the system with the fish.
Response: Due to the use of an automatic temperature-controlled aquaculture water circulation system, the type of heating device is unknown. In order to ensure the full growth of experimental fish, we often adopt a feeding method similar to the free feeding mode of livestock and poultry. For this reason, we try to offer the feed to fish by hand as much as possible at each meal until they have no intention of floating to the water surface. We used the standards (the feeding status of fish, appearance color of fish body, and examination of fish gills under dissecting microscope, etc.) to determine if fish are healthy.
Regarding the animal welfare of fish, we have provided them with adequate nutrition, proper water quality, behavioral freedom, and safety throughout the entire rearing experiment.
(16) Lines 119-212; are confusing: the number of fish anesthetized. Is there duplicate information?
Response: We have adjusted the description to avoid the confused expression.
(17) Line 124; were heparin syringes used? If so, this should be described.
Response: No, heparin was not used. This was done.
(18) Line 126; set the rpm to g.
Response: This was done.
(19) Lines 133-135; are confusing because they appear to indicate that all fish in the experiment were anesthetized. Verify the information.
Response: We have readjusted the description.
(20) Lines 151-157-162-164; mention the kit´s description.
Response: We have supplemented the kit instructions.
(21) Line 174; delete the period.
Response: This was done.
(23) Lines 207-209; this description regarding the way the results are represented raises questions. Why do the figures and the results description compare the 5 treatments? Clarify. If the data in Figures 1 through 4 are in the same figure, these data should be comparative with respect to the control and with each other.
Response: I understand what you tell. Because the FM group is only a negative control group relative to the high SBM group, there are significant differences between the two. The FM group can only be used to compare the intervention effect of glutamate on high SBM. To investigate the intervention effect of glutamate, it is necessary to examine the dose-response relationship of glutamate (SBM, G1, G2, and G3, that is, 0, 1, 2, and 3 glutamate levels) on the high SBM feed in order to have substantive significance. This practice has been applied in our team's previous research.
(24) Line 216; adjust the comment to the statistical comparison: "There are no differences between these 3 treatments."
Response: Thank you for your valuable comment. We adjusted the expression.
(25) In Table 3, how did mortality influence the calculations of feed consumed and FCR? If there was an adjustment, it should be mentioned.
Response: Considering the lower mortality rate and most of dead fish were at the early stage of the feeding duration, we did not adjust the feed consumed and FCR.
(26) Line 242; the letter r should be added.
Response: Thanks to your reminding. We added all the letter r to the place where it is needed.
(27) In Figures 1, 2, 3, and 4, the highest value should be marked with the letter a, the lowest value with the letter b, and so on.
Response: We adjusted the mark order.
(28) Line 255; is missing a parenthesis.
Response: This was done.
(29) Line 256; repeats the previous sentence.
Response: This was done.
(30) Figure captions 3 and 4, standardize them with the format of Figures 1 and 2.
Response: These were done.
(31) On Line 301; verify that the term "mucosal fold height" corresponds to the initials in Table 5 and is consistent with the description in the footnote.
Response: This was done to ensure consistency between the two.
(32) On Lines 301-302; verify that only two treatments, SBM and FM, are compared; however, the other treatments are not mentioned (with the same response as SBM).
Response: This was done. We have expanded the description.
(33) In Table 5, standardize the format in the footnotes.
Response: This was done to ensure consistency.
(34) In Figure 6, add the measurement bar in the corresponding unit.
Response: The measurement bar actually exists and the reason why it cannot be seen is that the image has been compressed. The scale bar is located at the bottom-left corner of each micrograph, but may appear fuzzy due to image compression.
(35) On Lines 328 and 332; standardize the P format of the document.
Response: This was done.
(36) On Line 330; add the letter r to each piece of data.
Response: This was done.
(37) On Line 336; standardize the footnotes with those of the other figures.
Response: This was done to ensure consistency.
(38) I suggest separating the discussion into sections based on the order of the results with their respective headings.
Response: Thank you for your encouraging comments. Originally, the discussion was structured to align sequentially with the results. Just as advocated by many scholars, the importance of focusing on metabolic interdependencies among the results is obvious. For this reason, we try our best to integrate different contents to emphasize the comprehensive influence of glutamate across all results and make it more compact in structure. Therefore, we abandoned the practice of arranging subheadings in the discussion section.
(39) On Line 456; verify that the journal appears as Fisheries and Aquatic Science
Response: This was done.

Reviewer 3 Report
Comments and Suggestions for Authors
The manuscript entitled «Glutamate supplementation ameliorated growth impairment and intestinal injury in high-soybean meal-fed orange-spotted groupers (Epinephelus coioides)» by Aozhuo Wang et al. explores how dietary glutamate can mitigate enteritis caused by soybean meal in aquaculture fish. Glutamate is known for its ability to promote intestinal health and its role in protein synthesis, while soya is rich in antinutritional factors. This study is significant for aquaculture as it aids in optimizing artificial fish feeds by incorporating cost-effective essential nutrients, such protein derived from soybean.
Basic reporting
Overall, the experiment was well-designed and yielded new data. The total panel of parameters explored to evaluate the physiology of fish growth – including the digestion and assimilation of feed (serum lipid profile and metabolite contents and metabolic and enzyme activities, intestinal hydrolysing enzyme activities and tissue ultrastructure, etc.) and inflammation (via inflammatory gene expression) – is both comprehensive and highly pertinent to the research objectives. The experimental design and methods are described in detail. The methodologies and statistical analyses used appear to be suitable and relevant to the study. A key finding is that glutamate supplementation may enhance the digestibility of plant-based feeds and reduce inflammatory responses, potentially mitigating growth retardation for aquaculture species. The conclusions drawn are aligned with the data collected.
As a non-native speaker, I am not qualified to judge the quality of the language though I feel the quality of English is satisfactory. Overall, the MS is consistent with the Animals topics and can be recommended for publication. Some minor comments detailed below need to be considered.
Minor comments
Line 14, please don’t introduce the abbreviation ‘ANFs’ in the Simple Summery (better in the main text) or decipher at the first mention aiming that the Simple Summary is a separate part of MS.
Table 1, lines 10 and 22, When describing a feed formulation, you mention ‘glutamate’ twice among the ingredients. I suggest indicating Glu content as ‘added Glu’ and ‘total Glu’ or otherwise.
Line 133, you write that the samples for intestinal microbiota analysis were taken though no methodology description and the obtained results have been given in the corresponding sections.
Line 160, the description of the parameters studied should be clarified, for example, glutathione (GSH) has no activity (you measured GSH level obviously); it is also not clear where GSH, T-AOC, CAT, and MDA have been determined – in the intestine tissue or in the content (semi-digested feed).
In summary, I recommend accepting the MS for publication with minor corrections.
Author Response
Reviewer 3
(1) The manuscript entitled «Glutamate supplementation ameliorated growth impairment and intestinal injury in high-soybean meal-fed orange-spotted groupers (Epinephelus coioides)» by Aozhuo Wang et al. explores how dietary glutamate can mitigate enteritis caused by soybean meal in aquaculture fish. Glutamate is known for its ability to promote intestinal health and its role in protein synthesis, while soya is rich in antinutritional factors. This study is significant for aquaculture as it aids in optimizing artificial fish feeds by incorporating cost-effective essential nutrients, such protein derived from soybean.
Response: Thanks to your encouraging comments.
(2) Basic reporting
Overall, the experiment was well-designed and yielded new data. The total panel of parameters explored to evaluate the physiology of fish growth – including the digestion and assimilation of feed (serum lipid profile and metabolite contents and metabolic and enzyme activities, intestinal hydrolysing enzyme activities and tissue ultrastructure, etc.) and inflammation (via inflammatory gene expression) – is both comprehensive and highly pertinent to the research objectives. The experimental design and methods are described in detail. The methodologies and statistical analyses used appear to be suitable and relevant to the study. A key finding is that glutamate supplementation may enhance the digestibility of plant-based feeds and reduce inflammatory responses, potentially mitigating growth retardation for aquaculture species. The conclusions drawn are aligned with the data collected.
Response: Thanks to your encouraging comments.
(3) As a non-native speaker, I am not qualified to judge the quality of the language though I feel the quality of English is satisfactory. Overall, the MS is consistent with the Animals topics and can be recommended for publication. Some minor comments detailed below need to be considered.
Response: Thanks to your encouraging comments.
(4) Minor comments
Line 14, please don’t introduce the abbreviation ‘ANFs’ in the Simple Summery (better in the main text) or decipher at the first mention aiming that the Simple Summary is a separate part of MS.
Response: We have made the change.
(5) Table 1, lines 10 and 22, When describing a feed formulation, you mention ‘glutamate’ twice among the ingredients. I suggest indicating Glu content as ‘added Glu’ and ‘total Glu’ or otherwise.
Response: Thank you for your valuable comments. We revised these.
(6) Line 133, you write that the samples for intestinal microbiota analysis were taken though no methodology description and the obtained results have been given in the corresponding sections.
Response: We have now deleted the mention of intestinal microbiota.
(7) Line 160, the description of the parameters studied should be clarified, for example, glutathione (GSH) has no activity (you measured GSH level obviously); it is also not clear where GSH, T-AOC, CAT, and MDA have been determined – in the intestine tissue or in the content (semi-digested feed).
Response: This was revised as suggested by the reviewer.
In summary, I recommend accepting the MS for publication with minor corrections.

Round 2
Reviewer 2 Report
Comments and Suggestions for Authors
Briefly, I would like to comment that the quality has improved, as most of the suggested changes to the document were reflected; however, I detect a few aspects that need to be revised. I attach the document in question after reviewing the manuscript.

Briefly, I leave the decision on the wording to the Editor-in-Chief. I see a few incomplete words in the document.
Author Response
Reviewer 2
(1) In the title, the introduction of the word Epinephelus coioides apparently requires a prior linking word.
Response: We set a link to introducing Epinephelus coioides in the introduction.
(2) In tables and figures, the order of the superscripts for statistical differences should be consistent, i.e., the highest value is a, the next lowest is b, and so on. This ensures that tables and figures match.
Response: We have readjusted the order of superscripts in tables to ensure their consistency match that of figures.
(3) In Table 1, I suggest using the units of chemical composition in g/kg, which is the correct order of units according to how the ingredients are listed.
Response: This was done.
(4) In Table 1, verify that the correct order is to list only Stay-C 35%.
Response: This was corrected.
(5) In Figure 5, the bar for structural or tissue measurements is still not clearly visible. I suggest verifying how this issue can be resolved.
Response: We readjusted the measurements to improve the resolution of the figure.
(6) On line 114, the word "initiation phase" has a different letter, and there is apparently a double space before the number 450.
Response: Thank you for your kind reminding. We removed an extra space.
(7) On line 22, there is apparently a double space before the letter a.
Response: Thank you for your kind reminding. We removed an extra space.
(8) On line 107, replace carbohydrate with carbohydrates.
Response: This was done.
(9) On line 136, the initials for survival rate are missing.
Response: This was done.
(10) On line 195, capitalize the x (100X).
Response: This was done.
(11) On line 197, verify that a version 3.0 appears online (verify). (https://www.cellularimaging.nl/wp-content/uploads/LASAF30_ReleaseNotes_confocal.pdf)
Response: We verify the release of Leica LAS AF software is v3.0 and inserted the link in the appropriate place.
Additionally, we have made a through paper polishing after completing the second round of revisions to the paper.
